# Bearing Fault Feature Extraction and Fault Diagnosis Method Based on Feature Fusion

**DOI:** 10.3390/s21072524

**Published:** 2021-04-04

**Authors:** Huibin Zhu, Zhangming He, Juhui Wei, Jiongqi Wang, Haiyin Zhou

**Affiliations:** 1College of Liberal Arts and Sciences, National University of Defense Technology, Changsha 410073, China; zhu_huibin@163.com (H.Z.); hzmnudt@163.com (Z.H.); wei_juhui@163.com (J.W.); wjq_gfkd@163.com (J.W.); 2Beijing Institute of Spacecraft System Engineering, China Academy of Space Technology, Beijing 100094, China

**Keywords:** bearing fault diagnosis, feature extraction, wavelet packet transform, singular value decomposition, entropy weight method, support vector machine

## Abstract

Bearing is one of the most important parts of rotating machinery with high failure rate, and its working state directly affects the performance of the entire equipment. Hence, it is of great significance to diagnose bearing faults, which can contribute to guaranteeing running stability and maintenance, thus promoting production efficiency and economic benefits. Usually, the bearing fault features are difficult to extract effectively, which results in low diagnosis performance. To solve the problem, this paper proposes a bearing fault feature extraction method and it establishes a bearing fault diagnosis method that is based on feature fusion. The basic idea of the method is as follows: firstly, the time-frequency feature of the bearing signal is extracted through Wavelet Packet Transform (WPT) to form the time-frequency characteristic matrix of the signal; secondly, the Multi-Weight Singular Value Decomposition (MWSVD) is constructed by singular value contribution rate and entropy weight. The features of the time-frequency feature matrix obtained by WPT are further extracted, and the features that are sensitive to fault in the time-frequency feature matrix are retained while the insensitive features are removed; finally, the extracted feature matrix is used as the input of the Support Vector Machine (SVM) classifier for bearing fault diagnosis. The proposed method is validated by data sets from the time-varying bearing data from the University of Ottawa and Case Western Reserve University Bearing Data Center. The results show that the algorithm can effectively diagnose the bearing under the steady-state and unsteady state. This paper proposes that the algorithm has better fault diagnosis capabilities and feature extraction capabilities when compared with methods that aree based on traditional feature technology.

## 1. Introduction

Rotating machinery is one of the most common classes of mechanical equipment and it plays a significant role in industrial applications [1]. As one of the key components in rotating machinery, bearings health directly affects the performance of mechanical equipment [2,3]. According to incomplete statistics, approximately 30% of failures are caused by the bearing fault [4]. Therefore, the fault diagnosis of bearing is of great significance for maintaining the safe operation of equipment [5].

Normally, it cannot be directly diagnosed due to the working environment of the bearing. Sensors can be used to collect digital signals that can reflect the state of the bearing [6,7,8,9,10], such as spectral signals [11], acoustic signals [12], and vibration signals. Spectral signals and acoustic signals can be used for non-destructive flaw detection, and have the advantages of obvious characteristic frequency and good early fault prediction. However, these methods require high professional quality of equipment and operators. The vibration signal of the bearing contains a wealth of fault energy information [13,14], and the collection of the bearing vibration signal does not require complex equipment and professionals. Therefore, fault diagnosis that is based on vibration signals is a common method for bearing diagnosis [15]. Vibration signals are affected by working conditions and equipment environment, the frequency spectrum is relatively complicated, and there are many interference factors. Therefore, the effective extraction of signal characteristics is the key to bearing fault diagnosis. The commonly used methods for extracting bearing signal features include empirical mode decomposition (EMD) and wavelet transform. EMD is an adaptive time-frequency analysis method without any prior knowledge, which has the ability of adaptive signal decomposition and noise reduction. However, EMD is only an empirical method and it lacks a complete theoretical basis [16]. Besides, in the decomposition process of EMD, modal aliasing is prone to occur due to problems, such as over-envelope, under-envelope, and unreasonable convergence conditions [17,18], which restricts the application of EMD. Wavelet packet transform (WPT) is a kind of wavelet transform. It can divide the frequency band of the signal into multiple scales to obtain information regarding signal in the low-frequency and high-frequency regions. Besides, WPT can adaptively select the corresponding frequency band to match the frequency spectrum of original signal according to the feature of signal, which has a more uniform frequency feature extraction effect [19,20]. Zhong et al. [21] used WPT to decompose the bearing signal, and the decomposed frequency band entropy is used as the input of Support Vector Machine (SVM) to establish a rolling bearing classification model.

The wavelet packet can extract the time-frequency information of the bearing vibration signal without omission, and more comprehensively describe the fault state of the bearing. However, on the one hand, it will increase the dimension of the bearing signal feature matrix and increase the computational complexity of the subsequent diagnosis model; on the other hand, there may be some insensitive features or even invalid features, increasing the probability that sensitive information will be submerged [22]. Therefore, after acquiring the bearing signal feature matrix, it is necessary to be further extracted to remove the irrelevant and redundant features. At present, the methods to remove redundant and irrelevant features of bearing include auto-encoder [23,24], neural networks [25,26], Principal Component Analysis (PCA) [27], kernel PCA [28], and Singular Value Decomposition (SVD). However, although intelligent algorithms, such as self-encoding and neural networks, have been applied to diagnose bearing faults, they have disadvantages, such as low generalization, slow calculation speed, and higher requirements for hardware equipment. For PCA and kernel PCA, on the one hand, PCA needs to be spatially transformed. Furthermore, the features of the original signal will lose their physical meaning through combination transformation; on the other hand, when using PCA, it is necessary to standardize the data. The noise in the data will affect the standardization process of data. SVD solves the dimensionality reduction order through the singular value of the matrix. When compared with PCA, the singular value has good stability and it is not sensitive to changes that are caused by interference, such as noise. It can still collect data information more accurately, even with small interference [29,30]. Kedadouche et al. [31] applied SVD to extract the matrix after WPT and use it as the input of SVM to identify the fault mode of rolling bearings. Cheng et al. [32] invented empirical mode decomposition to decompose the vibration signal of a rotating machine into multiple natural mode functions, and used SVD for the initial features matrix formed by these natural functions to obtain the singular values of matrix and used it for SVM fault diagnosis. Although SVD has good stability, as compared with PCA, the features extracted by SVD have relatively higher computational cost for subsequent diagnosis models. In view of this, Yuan et al. [33] proposed the Weighted Singular Value Decomposition (WSVD) with the ratio of singular values as the weight, and it is applied to radar emitter signals. The results showed that this method can extract the features of radar emitter signals very well. Although this method can effectively reduce the calculation cost of SVD, this method only tried to square the singular value after dimensionality reduction, which cannot fully reflect the information of the data itself and the importance of sensitive features.

This paper presents a study of the fault diagnosis method based on feature fusion when the bearing fault features are difficult to extract effectively which results in low diagnosis performance. Figure 1 shows the flowchart of the bearing fault diagnosis method based on feature fusion. The bearing vibration signal that is collected by the sensor obtains the time-frequency domain characteristics of the bearing through wavelet packet transform (WPT). This time-frequency domain feature is reduced dimension by the Multi-Weight Singular Value Decomposition (MWSVD). The reduced dimensionality features are used in SVM for fault diagnosis. The experimental results show the superiority of this method when compared to some of the traditional feature techniques. The major contributions of this paper include the following:

(1) a feature extraction method that is based on MWSVD is proposed and its effectiveness in two data set is evaluated. In the proposed method, the time-frequency domain information of the vibration signal that is extracted by WPT is best preserved in the low-dimensional space;

(2) the algorithm proposed in this paper is compared with some traditional feature extraction algorithms, combined with support vector machines for fault diagnosis, and the diagnosis effect is compared; and,

(3) a bearing fault diagnosis algorithm based on feature fusion is proposed, which can timely and effectively diagnose bearings in both steady state and non-steady state.

The rest of this article is as follows: Section 2 introduces the Weighted Singular Value Decomposition (WSVD) algorithm. Section 3 describes the process of wavelet packet decomposition and weighted singular value decomposition, and it proposes a feature extraction method based on fusion multi-weight singular value decomposition. Section 4 proposes a fault diagnosis method that is based on feature fusion. Section 5 shows the fault diagnosis results of the two data sets and the comparison results with other methods. Section 6 draws the conclusion.

## 2. Weighted Singular Value Decomposition Method

The principle and steps of weighted singular value decomposition are as follows [33]:

Firstly, the data can be normalized by
(1)$${\hat{A}}_{i}=\frac{2\times (A_{i}-{\bar{A}}_{i})}{|\max\left(A_{i}\right)-\min\left(A_{i}\right)|}$$
where $A_{i}$ is the *i*th row data of matrix *A*, ${\hat{A}}_{i}$ is the *i*th row data after data normalization, and ${\bar{A}}_{i}$ is the mean value of $A_{i}$. Secondly, perform SVD decomposition according to the following equation
(2)$$\hat{A}=U_{m\times m}\Sigma_{m\times s}V_{s\times s}^{T}$$
where $\hat{A}$ is the normalized matrix of *A*, $\Sigma_{m\times s}={\left[\Lambda_{s\times s}\;\;\;\;\;\;O_{(m-s)\times s}^{T}\right]}^{T}$, $O_{(m-s)\times s}$ is zero matrix, $\Lambda_{s\times s}=diag\left(\sigma_{1},\sigma_{2},\cdots ,\sigma_{s}\right)$, $\sigma_{1}\geq \sigma_{2}\geq \cdots \geq \sigma_{s}$ is singular value, $U_{m\times m}$ and $V_{s\times s}$ are the unitary matrix. The order r<s after dimensionality reduction is determined by the cumulative contribution rate of singular value that is greater than 90%. Subsequently, $\Sigma_{m\times s}$ becomes $\Sigma_{r\times r}=diag\left(\sigma_{1},\sigma_{2},\cdots ,\sigma_{r}\right)$,$\sigma_{1}\geq \sigma_{2}\geq \cdots \geq \sigma_{r}$ after dimensionality reduction. The weight is calculated according to the elements in $\Sigma_{r\times r}$
(3)wi=σiσi∑i=1rσi∑i=1rσi, =1, 2…r

Let the weight vector be ${\left[w_{i}\right]}_{1\times r}=\left[w_{1},\cdots ,w_{r}\right]$, according to the weight
(4)$$D_{m\times r}=\left|U_{m\times r}\right|⊗w_{i}=\left(\begin{array}{ccc} \left|u_{11}\right|w_{1} & \ldots & \left|u_{1r}\right|w_{r}\\ \vdots & \ddots & \vdots \\ \left|u_{m1}\right|w_{1} & \cdots & \left|u_{mr}\right|w_{r} \end{array}\right)$$
where $u_{ij}$ is the element of row *i* and column *j* of the matrix $U_{m\times r}$. The matrix $D_{m\times r}$ can be normalized according to the following equation to obtain the weighted singular value decomposition matrix *B*
(5)$$B=\left(\begin{array}{ccc} d_{11}/\zeta_{1} & \ldots & d_{1r}/\zeta_{1}\\ \vdots & \ddots & \vdots \\ d_{1m}/\zeta_{m} & \cdots & d_{mr}/\zeta_{m} \end{array}\right)$$
where $\zeta_{i}=\sum_{j}^{r}d_{ij},i=1,\cdots ,m$, $d_{ij}$ is the element in row *i* and column *j* of the matrix $D_{m\times r}$.

## 3. Fault Feature Extraction Method Based on WPT-MWSVD

### 3.1. WPT

Suppose that Z is the set of integers, $L^{2}\left(R\right)$ is a square-integrable real function space, and a series of closed subspace sequence ${\left\{V_{l}\right\}}_{l\in Z}$ on $L^{2}\left(R\right)$ is called the multi-resolution analysis of space $L^{2}\left(R\right)$ if the following conditions are met:

(**1**) Monotonicity: $V_{l+1}\subset V_{l},l\in Z$;

(**2**) Translation invariance: $f\left(x\right)\in V_{l}\Leftrightarrow f\left(x-\alpha \right)\in V_{l},\alpha \in Z$;

(**3**) Scalability: $f\left(x\right)\in V_{l}\Leftrightarrow f\left(2x\right)\in V_{l-1}$;

(**4**) Approximation: $\bar{\underset{l\in Z}{\cup }V_{l}}=L^{2}\left(R\right),\underset{l\in Z}{\cap }V_{l}=\left\{0\right\}$;

(**5**) The existence of Riesz base: {φ(x−k)|k∈Z} form the Risez base of $V_{0}$.

If {φ(x−k)|k∈Z} is a canonical orthonormal basis of $V_{0}$ in ${\left\{V_{l}\right\}}_{l\in Z}$ for multi-resolution analysis of $L^{2}\left(R\right)$, then $\{\varphi_{l,k}\left(x\right)=2^{-\frac{l}{2}}\varphi \left(2^{-l}x-k\right){\left|k\in Z\right\}}_{l\in Z}$ is the canonical orthonormal basis of ${\left\{V_{l}\right\}}_{l\in Z}$ namely $V_{l}=span{\left\{2^{-\frac{l}{2}}\varphi \left(2^{-l}x-k\right)\right\}}_{k\in Z}$ [34].

Notice that $V_{l+1}\subset V_{l}$, if the orthogonal complement of $V_{l+1}$ in $V_{l}$ is $W_{l+1}$, that is $V_{l}=V_{l+1}⊕W_{l+1}$, then $W_{l+1}$ is called the wavelet subspace of $L^{2}\left(R\right)$

If $W_{0}$ is a orthogonal complement of $V_{0}$ on $V_{-1}$, then $\{\psi_{l,k}\left(x\right)=2^{-\frac{l}{2}}\psi \left(2^{-l}x-k\right){\left|k\in Z\right\}}_{l\in Z}$ is the orthogonal basis of $W_{l}$, that is $W_{l}=span{\left\{2^{-\frac{l}{2}}\psi \left(2^{-l}x-k\right)\right\}}_{k\in Z}$ [35].

Suppose that the bearing signal f(x) belongs to $V_{l}$, WPT can decompose f(x) in the form of a binary tree. The principle of WPT can be described, as follows [34].

Suppose that ${\left\{V_{l}\right\}}_{l\in Z}$ is a multi-resolution analysis of $L^{2}\left(R\right)$, φ(x) and ψ(x) are the corresponding orthogonal scaling function and orthogonal wavelet function, and the two-scale equations are satisfied
(6)$$\left\{\begin{array}{c} \varphi \left(x\right)=\sqrt{2}\sum_{k\in Z}h_{k}\varphi \left(2x-k\right)\\ \psi \left(x\right)=\sqrt{2}\sum_{k\in Z}g_{k}\varphi \left(2x-k\right) \end{array}\right.$$
where $h_{k}\left(\cdot \right)$ and $g_{k}\left(\cdot \right)$ are low-pass and high-pass filters, respectively. Let $\mu_{0}=\varphi \left(x\right),\mu_{1}=\psi \left(x\right)$, then
(7)$$\left\{\begin{array}{c} \mu_{0}\left(x\right)=\sqrt{2}\sum_{k\in Z}h_{k}\mu_{0}\left(2x-k\right)\\ \mu_{1}\left(x\right)=\sqrt{2}\sum_{k\in Z}g_{k}\mu_{0}\left(2x-k\right) \end{array}\right.$$

The above formula is extended to the general situation
(8)$$\left\{\begin{array}{c} \mu_{2n}\left(x\right)=\sqrt{2}\sum_{k\in Z}h_{k}\mu_{n}\left(2x-k\right)\\ \mu_{2n+1}\left(x\right)=\sqrt{2}\sum_{k\in Z}g_{k}\mu_{n}\left(2x-k\right) \end{array}\right.$$

From Equation (Equation 8), the function set $\left\{\mu_{n}\left(t\right):n=0,1,2,\cdots \right\}$ can be obtained that is called wavelet packet determined by the orthogonal scaling function φ(x). The corresponding space of wavelet packet $\left\{\mu_{n}\left(x\right):n=0,1,2,\cdots \right\}$ is
(9)$$U_{l}^{n}={\bar{span\{2^{-\frac{l}{2}}\mu_{n}\left(2^{-l}x-k\right),k\in Z\}}}_{l\in Z}$$

Thus, the following formula is established [34]
(10)$$\begin{array}{c} \sqrt{2}\mu_{n}\left(2^{-l}x-k\right)=\sum_{b\in Z}{\bar{h}}_{k-2b}\mu_{2n}\left(2^{-l-1}x-b\right)\\ \;\;\;\;\;\;\;\;\;\;\;\;\;\;\;\;\;\;\;\;\;\;\;\;\;\;\;\;\;\;\;\;\;\;\;\;\;\;\;\;\;\;\;\;\;\;\;\;\;\;\;\;\;\;\;\;\;\;\;\;\;\;\;\;+\sum_{b\in Z}{\bar{g}}_{k-2b}\mu_{2n+1}\left(2^{-l-1}x-b\right),k\in Z \end{array}$$
where $\bar{h}\left(\cdot \right)$ and $\bar{g}\left(\cdot \right)$ are the complex conjugates of $h_{k}\left(\cdot \right)$ and $g_{k}\left(\cdot \right)$. According to $V_{\gamma }=V_{\gamma +1}⊕W_{\gamma +1}$, it can be obtained that
(11)$$U_{l}^{n}=U_{l+1}^{2n}⊕U_{l+1}^{2n+1},l\in Z$$

In the case of l=3, the corresponding structural decomposition is shown in Figure 2. Suppose that $f_{l}^{n}\left(x\right)\in U_{l}^{n}$, $f_{l+1}^{2n}\left(x\right)\in U_{l+1}^{2n}$, $f_{l+1}^{2n+1}\left(x\right)\in U_{l+1}^{2n+1}$, then
(12)$$\begin{array}{c} f_{l}^{n}\left(x\right)=\sum_{k\in Z}c_{l,k}^{n}2^{-\frac{l}{2}}\mu_{n}\left(2^{-l}x-k\right)\;\;=f_{l+1}^{2n}\left(x\right)+f_{l+1}^{2n+1}\left(x\right)\\ \;\;\;\;\;\;\;\;\;\;\;\;\;\;\;\;\;\;\;\;\;\;\;\;\;\;\;\;\;\;\;=\sum_{b\in Z}c_{l+1,b}^{2n}2^{-\frac{l+1}{2}}\mu_{2n}\left(2^{-l-1}x-b\right)\;+\sum_{b\in Z}c_{l+1,b}^{2n+1}2^{-\frac{l+1}{2}}\mu_{2n+1}\left(2^{-l-1}x-b\right) \end{array}$$
where $c_{l,k}^{n}$, $c_{l+1,b}^{2n}$ and $c_{l+1,b}^{2n+1}$ are the coefficients of function $f_{l}^{n}\left(x\right)$, $f_{l+1}^{2n}\left(x\right)$, $f_{l+1}^{2n+1}\left(x\right)$ under the corresponding subspace bases. By substituting Equation (Equation 10) into Equation (Equation 12), it is concluded that   
(13)$$\left\{\begin{array}{c} c_{l+1,b}^{2n}=\sum_{k\in Z}c_{l,k}^{n}{\bar{h}}_{k-2b},b\in Z\\ c_{l+1,b}^{2n+1}=\sum_{k\in Z}c_{l.k}^{n}{\bar{g}}_{k-2b},b\in Z \end{array}\right.$$

Equation (Equation 13) is called the Mallat decomposition algorithm formula of wavelet packet [36]. In application, for the bearing continuous signal f(x), the sample sequence f(t),t=1,2,⋯,mλ that is obtained by sampling can be directly approximated, as follows
(14)$$c_{0,\lambda }^{0}=f\left(t\right),t=1,2,\cdots ,m\lambda$$
where mλ is the sampling length of the signal. Therefore, as long as the type of wavelet packet function and scales *l* are selected, all of the wavelet packet coefficients $c_{l,\nu }^{n}$ of bearing signal sequence f(t),t=1,2,⋯,mλ under the scales *l* are obtained by Mallat decomposition algorithm formula, where $\nu =1,2,\cdots ,m\lambda /2^{l}\;$, $n=0,1,\cdots ,2^{l}-1$ is the number of nodes corresponding to the scales *l* [37]. Taking the scale l=3 as an example, Figure 3 shows the corresponding Mallat decomposition process. The characteristic matrix $A_{m\times s}=\left(a_{ij}\right)$ of bearing signal is constructed by wavelet packet coefficient $c_{l,\nu }^{n}$, where *m* is the number of samples of bearing signal, λ is the length of a single sample, and $s=2^{l}$ is the number of all wavelet coefficients of a single sample at the scale *l*. The element $a_{ij}$ in the matrix $A_{m\times s}$ is the *j*-th wavelet packet coefficient energy at the scale *l* that is obtained by the *i*-th sample through WPT. Algorithm 1 and Figure 4 show the specific process.
**Algorithm 1** Wavelet packet decomposition of bearing signal**Input:** bearing signal sequence f(t),t=1,2,⋯,mλ, window width λ, wavelet packet scale *l*, wavelet packet function type**Output:** Time-frequency feature matrix $A_{m\times s}$1:Perform sliding window processing on the bearing signal sequence f(t),t=1,2,⋯,mλ;2:The f(t),t=1,2,⋯mλ was divided into *m* sequence, and each sequence fragment was λ;3:**for**j=1:m**do**4:  According to the type of wavelet packet function, the wavelet packet coefficient $c_{l,\nu }^{n}$ of the *j*-th sample is obtained by Mallat decomposition algorithm formula of wavelet packet;5:  $c_{l,\nu }^{n}$ is arranged according to the corresponding order under the *l*-th scale to form the *j*-th row of $A_{m\times s}$;6:**end for**

Suppose that the sequence f(t),t=1,2,⋯,mλ is divided into *m* fragments, each fragment is λ, as shown in Figure 3 and Figure 4.

### 3.2. Multiple Weighted Singular Value Decomposition Method

The time-frequency matrix $A_{m\times s}$ that is obtained by WPT contains some insensitive features. This paper proposes a multi-weight singular value decomposition algorithm based on WSD in order to effectively extract the sensitive information in the time-frequency feature matrix and eliminate the correlation between variables.

Firstly, the feature matrix $A_{m\times s}$ can be normalized by
(15)$$A_{m\times s}^{*}=\frac{A_{m\times s}-{\bar{A}}_{m\times s}}{Var{\left(A_{m\times s}\right)}^{1/2}}$$
where $A_{m\times s}^{*}$ is the normalized matrix of $A_{m\times s}$, ${\bar{A}}_{m\times s}$ is the mean of $A_{m\times s}$, $Var{\left(A_{m\times s}\right)}^{1/2}$ is the standard deviation of $A_{m\times s}$. Similar to the WSVD algorithm, the singular value decomposition of the matrix $A_{m\times s}^{*}$ is performed according to Equation (Equation 2) to obtain $U_{m\times m}$, $\Sigma_{m\times s}$ and $V_{s\times s}$. Because the characteristic matrix $A_{m\times s}$ is the projection coefficient of the sample sequence f(t),t=1,2,⋯,mλ of the bearing signal on the wavelet packet subspace, the matrix $A_{m\times s}$ is a real matrix, $U_{m\times m}$ and $V_{s\times s}$ is orthogonal matrices. Similar to the WSVD algorithm, the order r<s after dimension reduction is determined by the cumulative contribution rate of singular value. Subsequently, $\Sigma_{m\times s}$ becomes $\Sigma_{r\times r}$ after dimensionality reduction. The feature matrix $D_{m\times s}$ after the first weighting is calculated, as follows
(16)$$D_{m\times r}=U_{m\times r}\times \Sigma_{r\times r}⊗w_{i}=\left(\begin{array}{ccc} u_{11}\sigma_{1}w_{1} & \ldots & u_{1r}\sigma_{r}w_{r}\\ \vdots & \ddots & \vdots \\ u_{m1}\sigma_{1}w_{1} & \cdots & u_{mr}\sigma_{r}w_{r} \end{array}\right)$$

The second weighted weight is obtained by the idea of information entropy. The feature matrix $D_{m\times s}$ needs to be processed according to the following formula before calculating the entropy value
(17)$$d_{ij}^{*}=\frac{d_{ij}-\min\left\{d_{1j},\cdots ,d_{mj}\right\}}{\max\left\{d_{1j},\cdots ,d_{mj}\right\}-\min\left\{d_{1j},\cdots ,d_{mj}\right\}},i=1,\cdots ,m;j=1,\cdots ,r$$
where $d_{ij}$ is the element of row *i* and column *j* of the matrix $U_{m\times r}$. $d_{ij}^{*}$ is the element of row *i* and column *j* of the matrix $D_{m\times r}^{*}$. The information entropy of the matrix $H_{j},j=1,2,\cdots ,r$ is calculated, as follows
(18)Hj=−∑i=1mpijlnpij−∑i=1mpijlnpijlnmlnmpij=dij*dij*∑i=1mdij*∑i=1mdij*i=1,2,⋯m;j=1,2,⋯r

if $p_{ij}=0$, $\lim_{p_{ij}\to 0}\;p_{ij}\cdot \ln p_{ij}=0$. According to the information entropy of the matrix $H_{j},j=1,2,\cdots ,r$, the entropy weight is calculated, as follows:(19)$$\varpi_{j}=\frac{1-H_{j}}{r-\sum H_{j}}\;\;,\;\;j=1,2,\cdots ,r$$

The weighted characteristic matrix is defined using Equation (Equation 15)
(20)$$T_{m\times r}=D_{m\times r}^{*}⊗\varpi_{j}=\left(\begin{array}{ccc} d_{11}^{*}\varpi_{1} & \ldots & d_{1k}^{*}\varpi_{r}\\ \vdots & \ddots & \vdots \\ d_{m1}^{*}\varpi_{1} & \cdots & d_{mk}^{*}\varpi_{r} \end{array}\right)$$

Figure 5 shows the specific process of MWSVD.

In conclusion, the feature extraction method of bearing fault is given, as follows Algorithm 2.
**Algorithm 2** Feature extraction method of bearing fault**Input:** bearing signal sequence f(t),t=1,2,⋯,mλ, window width λ, wavelet packet scale *l*, wavelet packet function type**Output:** The feature matrix $T_{m\times r}$1:The time-frequency feature matrix $A_{m\times s}$ is obtained by Algorithm 1;2:The time-frequency matrix $A_{m\times s}$ is normalized. SVD is decomposed according to Equation (Equation 2), and weight is calculated according to Equation (Equation 3);3:The matrix $D_{m\times r}$ is obtained according to Equation (Equation 16), the matrix $D_{m\times r}^{*}$ is obtained according to Equation (Equation 17);4:The entropy weight of the matrix $D_{m\times r}^{*}$ is obtained according to Equation (Equation 18);5:The characteristic matrix $T_{m\times r}$ of the bearing fault is obtained according to Equation (Equation 20).

## 4. Fault Diagnosis Method Based on Feature Fusion

The core idea of SVM is to transform indivisible samples in low-dimensional space into high-dimensional space through a kernel function, and realize the classification between samples by seeking the optimal classification hyperplane [38].

Suppose that sample set $\left\{\left(x_{i},y_{i}\right)|,x_{i}\in R^{n},y_{i}\in \left\{-1,+1\right\},i=1,2,\cdots ,q\right\}$, where *q* is the number of training samples, and $x_{i}$ and $y_{i}$ are the *i*-th data points that belong to a binary class $y_{i}$.

SVM maps the input of the low-dimensional space to the high-dimensional space by the nonlinear mapping θ(⋅) to obtain the linear classification function $f\left(x\right)=\omega^{T}\theta \left(x\right)+b$, where ω is the weight and *b* is the offset. For a binary classification issue with labels −1 and 1, all of the samples should meet a specific condition, as defined in Equation (Equation 21), thus the two types of samples can be completely separated:(21)$$f\left(x\right)=w^{T}\theta \left(x\right)+b\left\{\begin{array}{c} >1\;\;\;\;\;\;\;\;\;\;\;\;\;\;\;\;\;\;\;\;\;\;for\;\;\;\;\;\;\;y_{i}=1\\ <-1\;\;\;\;\;\;\;\;\;\;\;\;\;\;for\;\;\;\;\;\;\;\;y_{i}=-1 \end{array}\right.$$

To linearly solve non-separable problems, slack variable $\xi_{i}$ and penalty factor *C* are introduced, thus the best classification function is obtained by solving the minimum value of Equation (Equation 22)
(22)$$\frac{1}{2}\left|\right|\omega {\left|\right|}_{2}^{2}+C\sum_{i=1}^{N}\xi_{i}$$

The Lagrange coefficient is introduced, Equation (Equation 22) is transformed into a quadratic programming problem to solve
(23)$$L\left(\alpha \right)=\sum_{i=1}^{N}\alpha_{i}-\frac{1}{2}\sum_{i,j=1}^{N}\alpha_{i}\alpha_{j}y_{i}y_{j}K\left(x_{i},x_{j}\right)$$
where $K(x_{i},x_{j})$ is the kernel function. By solving the smallest L(α), the final classification function is as follows
(24)$$f\left(x\right)=\mathrm{sgn}(\sum_{i,j=1}^{N}\alpha_{i}y_{i}K\left(x_{i},x_{j}\right)+b)$$

This paper chooses the Gaussian kernel function as the kernel function of SVM. Its expression is as follows
(25)$$K\left(x_{i},x_{j}\right)=\exp\left(-\frac{|x_{i}-x_{j}|}{2\varepsilon^{2}}\right)$$
where ε is the kernel parameter. The penalty parameter *C* and the kernel parameter ε have an important influence on the classification accuracy and generalization ability. There is currently no unified theoretical method to find the best combination of the above two parameters. This paper uses the genetic algorithm to find the optimal value of the parameter

This paper uses the following equation to calculate the classification accuracy η of SVM
(26)η=∑j=1Sβj∑j=1SβjSS,βj=1ifpredictinglabel=actuallabel0ifpredictinglabel≠actuallabel
where *S* is the number of samples in the test set. The appeal fault diagnostic model was run τ times and the variance δ of the classification accuracy η is calculated, as follows
(27)$$\left\{\begin{array}{c} \delta =\sum_{i=1}^{\tau }\frac{{\left(\eta_{i}-\bar{\eta }\right)}^{2}}{\tau -1}\\ \bar{\eta }=\sum_{i=1}^{\tau }\frac{\eta_{i}}{\tau } \end{array}\right.$$

In conclusion, the bearing fault diagnosis method based on feature fusion is proposed

## 5. Experiments and Analysis Results

### 5.1. Case A: The Time-Varying Bearing Data from the University of Ottawa

The time-varying bearing data from the University of Ottawa [39]. The experiments are performed on a SpectraQuest machinery fault simulator (MFS-PK5M). The experimental set-up is shown in Figure 6. The shaft is driven by a motor and the rotational speed is controlled by an AC drive. Two ER16K ball bearings are installed to support the shaft, the left one is a healthy bearing and the right one is the experimental bearing, which are replaced by bearings of different health conditions. An accelerometer (ICP accelerometer, Model 623C01) is placed on the housing of the experimental bearing to collect the vibration data. In addition, an incremental encoder (EPC model 775) is installed to measure the shaft rotational speed. To ensure the authenticity of the data, three trials are collected for each experimental setting. In this article, the operating speed condition selected is deceleration. Table 1 shows the operating speed and health status of the selected bearing.

In this paper, Table 1 shows each fault state and its corresponding label. Signals are sampled at 200 KHz. For each state, 76,800 points are collected and labelde in turn. There are also some other research data in the data set, which are not described because they are not used in this article. Table 2 shows the experimental environment and experimental parameters of this article.

According to Algorithm 3, 60 groups are randomly selected from the state category of each bearing as the training set, and 60 groups are used as the test set, which are labeled according to the state category that they belong to. To illustrate the effectiveness of the proposed method, PCA, SVD, and WSVD [30] are selected as the comparison, Furthermore, the SVM classifier is obtained from the training set data, and the classification accuracy and diagnosis time of the test set data by the SVM classifier are used as the criteria for assessing the optimal diagnosis method. To further illustrate the effectiveness of MWSVD method that is proposed in this paper, after feature extraction three feature extraction methods are visualized and analyzed to observe the effects of feature extraction.
**Algorithm 3** Bearing fault diagnosis method based on Feature Fusion**Input:** bearing signal sequence f(t),t=1,2,⋯,mλ, window width λ, wavelet packet scale *l*, wavelet packet function type, Number of runs τ.**Output:** classification accuracy η, calculation time, Variance of classification accuracy δ.1:The feature matrix of bearing fault is obtained $T_{m\times r}$ by Algorithm 2;2:The feature matrix $T_{m\times r}$ is randomly divided into the training set and test set, and different state types are labeled;3:The SVM classifier is trained by the training set to obtain the SVM-based classification model;4:The test set is input to the SVM-based classification model to obtain the predicted label of the test set. The actual label and predicted label of the test set are calculated according to Equation (Equation 26) to calculate the classification accuracy η of the diagnostic model. The total running time of bearing signal from MVSVD feature extraction to training SVM classification model to test result is calculated;5:The model is run τ times in sequence to get the classification accuracy η each time, and the variance δ of classification accuracy is obtained according to Equation (Equation 27).

In this paper, a genetic algorithm is used to find the optimal parameters of SVM in the training set after five-fold cross-validation. Table 3 shows the optimal parameters and classification results of SVM on the training set under the optimal parameters (CA accuray).

Figure 7 is the Receiver Operating Characteristic Curve (ROC) curve diagram of the four algorithms on bearing fault diagnosis in a single experiment. It shows that WPT-MWSVD+SVM can effectively diagnose the fault of the bearing inner race and outer race as compared to the other three methods. Figure 8 shows the Classification confusion matrix of four algorithms on bearing fault diagnosis in a single experiment. It can be seen that: WPT-WSVD+SVM and WPT-MWSVD+SVM can effectively distinguish the normal state of the bearing; WPT-MWSVD+SVM can effectively diagnose the inner race fault, and the diagnosis effect is better than the other three methods; WPT-SVD+SVM and WPT-MWSVD+SVM can both effectively distinguish outer race faults, and the diagnostic effect is much better than the other two methods. The four algorithms are run for 100 times in sequence, and the experimental results are shown in Figure 9 and Figure 10, Table 4. It can be seen that the average classification accuracy of this method is 87.87%, which is higher than the other three methods, and the average time used is 16.32 s, which is significantly lower than the other three methods. This shows that the proposed method has better computational efficiency and diagnostic accuracy. Besides, it can be seen from Figure Figure 9b that the fluctuation of the classification accuracy of this method is small. Table 4 shows the variance of classification accuracy. To examine the computational cost of this method in analyzing experimental data, refer to reference [42], and calculate the computational efficiency of the model based on the classification time; the related processing times are listed in Table 4. The results show that the average diagnosis time for WPT-MWSVD+SVM model diagnosis to collect 1 s sample data only takes 10.63 s. Therefore, the method proposed in this paper is superior to the other three methods for bearing fault diagnosis. Because the difference between the four fault diagnosis methods lies on the extraction of bearing fault features, this shows that the MWSVD feature extraction method proposed in this paper can effectively extract sensitive features of bearing information and it has good feature extraction capabilities.

The four methods are visualized and analyzed to further illustrate the feature extraction capability of MWSVD, which are shown in Figure 11. It can be seen that the number of principal components extracted by PCA is less than the number of singular values extracted by the other three methods under the same cumulative singular value contribution rate. WPT-MWSVD has a more scattered distribution of data samples as compared with the other three methods, which can not only effectively improve the classification accuracy of subsequent fault diagnosis, but also effectively shorten the fault diagnosis time, which corresponds to the results presented in Table 4. Therefore, the MWSVD method that is constructed in this paper can effectively extract bearing signal features and improve the classification ability of SVM classifier.

### 5.2. Case B: Case Western Reserve University Bearing Data Set

The data used in this case are taken frome the Case Western Reserve University Bearing Data Set [43]. Figure 12 presents a schematic diagram of the experimental platform, the test stand consists of a 2 hp motor (left), a torque transducer/encoder (center), a dynamometer (right), and control electronics (not shown). The test bearings support the motor shaft. Vibration data were collected using accelerometers, which were attached to the housing with magnetic bases. The test bearing are SKF6205-2RS deep groove ball bearing.

In this article, we choose the vibration acceleration signals were collected under the condition that rotor speeds 1730 r/min with sampling frequency 12 kHz. These data include three fault levels and four fault states. For each state under each fault level, 76,800 points are collected and labeled in turn. Table 5 presents the details.

In this case, the wavelet basis function we choose is db6, and the number of wavelet layers is l=3. The experimental environment, experimental parameters, selection of training set and test set, selection of comparison model, and evaluation criteria of Section 5.2 are the same as Section 5.1, except that wavelet function, wavelet packet scale, and the selected data set.

In this paper, a genetic algorithm is used to find the optimal parameters of SVM in the training set after five-fold cross-validation. Table 6 shows the optimal parameters and classification results of SVM on the training set under the optimal parameters (CA accuracy).

Figure 13 presents the ROC curve diagram of the four algorithms on bearing fault diagnosis in a single experiment. It can be seen that Minor fault: the ROC curve area of the other three methods is larger than that of WPT-WSVD+SVM under inner race fault and outer race fault, and the ROC curve area of WPT-MWSVD+SVM is larger than the other three methods under ball fault; general failure: the curve areas of WPT-SVD+SVM and WPT-MWSVD+SVM are larger than the other two methods under three types of failures; serious failures: the ROC curve area of WPT-MWSVD+SVM is larger than the other three methods under inner race fault and outer race fault. This shows that WPT-MWSVD+SVM has better bearing fault diagnosis capabilities. Figure 14 is the classification confusion matrix of four kinds of bearing fault diagnosis in a single experiment under different fault degrees. It can be seen that Minor fault: the diagnostic ability of WPT-MWSVD+SVM is better than the other three methods under ball fault; general failure: the diagnostic capabilities of WPT-WSVD+SVM and WPT-MWSVD+SVM are much better than the other two methods under inner race fault and ball fault; serious failures: the diagnostic capabilities of WPT-WSVD+SVM and WPT-MWSVD+SVM are better than the other two methods under inner race fault. WPT-MWSVD+SVM can effectively diagnose outer race fault, and the diagnostic effect is better than the other three methods. The four algorithms are run 100 times, in turn, and the experimental results are shown in Figure 15 and Figure 16 and Table 7. The results show that this method has the advantages of high classification accuracy and short calculation time under the three failure levels. Besides, in Figure 15, the fluctuation of the classification accuracy of this method is small. It shows that the algorithm in this paper can enhance the sensitive features of bearing signals and reduce the interference of insensitive features on the diagnosis model after twice weighting. Therefore, the diagnosis model that is proposed in this paper has great accuracy. Table 7 lists the related processing times. The results show that the average diagnosis time for WPT-MWSVD+SVM model diagnosis to collect 1 s sample data only takes 10.63 s. Therefore, the method that is proposed in this paper is superior to the other three methods for bearing fault diagnosis.

The four methods are visualized and analyzed to further illustrate the feature extraction capability of the MWSVD method constructed in this paper, and the results are shown in Figure 17, Figure 18 and Figure 19. It can be seen that the number of principal components extracted by PCA is less than the number of singular values extracted by the other three methods under the same cumulative singular value contribution rate. WPT-MWSVD has a more scattered distribution of data samples as compared with the other three methods, which can not only effectively improve the classification accuracy of subsequent fault diagnosis, but also effectively shorten the fault diagnosis time, which corresponds to the results shown in Table 7. Therefore, the MWSVD method that is constructed in this paper can effectively extract bearing signal features and improve the classification ability of SVM classifier.

To sum up, we can see that:(1)In different bearing data sets or different failure degrees, the four algorithms are run 100 times in sequence. The bearing fault diagnosis method that is based on feature fusion proposed in this paper has a high average classification accuracy rate. This model has a shorter average time than the other three fault diagnosis methods, and the average diagnosis time for the model diagnosis to collect 1-s sample data is the lowest. This shows that the method in this paper not only has higher accuracy, but also lower computational cost in bearing fault diagnosis; and,(2)In different bearing data sets or different failure degrees, the four feature extraction algorithms are visualized and analyzed. The results show that, as compared with the traditional feature extraction methods, the MWSVD feature extraction method proposed in this paper can retain more bearing signals information. Besides, the feature distribution of bearing signal extracted in this paper is relatively divergent. This means that the MWSVD feature extraction method proposed in this paper can effectively extract bearing signal features, reduce the computational complexity of subsequent diagnostic models, and improve the diagnostic capabilities of subsequent diagnostic models.

## 6. Conclusions

To cope with the problem that it is difficult to extract feature vector effectively in rolling bearing fault diagnosis, our work is as follows: firstly, this paper constructs an SVD feature extraction method thatis based on the fusion of multiple weights through the contribution rate of singular values and entropy weights. On the one hand, this method makes up for the problem that the traditional PCA algorithm loses its physical meaning due to the combination transformation of features in the process of removing feature redundancy, and it reduces the impact of noise on the data; on the other hand, it makes up for the problem that the effects of the features extracted by the traditional SVD algorithm have a high computational cost for subsequent models. Secondly, this paper combines it with the SVM classifier to propose a bearing fault diagnosis method that is based on feature fusion. Finally, the time-varying bearing data of the University of Ottawa and the data set of Case Western Reserve University bearing data center are used in the experiment. It shows that, under the condition of the steady-state and non-steady-state of bearing, under different sampling frequency and sampling time of bearing signal, and under a different degree of damage of bearing, MWSVD can effectively extract the sensitive features in the bearing and reduce the interference of non-sensitive features to the diagnosis model. WPT-MWSVD+SVM diagnosis models can quickly and accurately identify bearing faults, have good model adaptability, high calculation accuracy and calculation efficiency, and they have great application potential. Besides, the SVD-based MWSVD feature extraction algorithm is also suitable for other aspects of dimensionality reduction requirements, which will be the author’s next research direction.

## Figures and Tables

**Figure 1 sensors-21-02524-f001:**
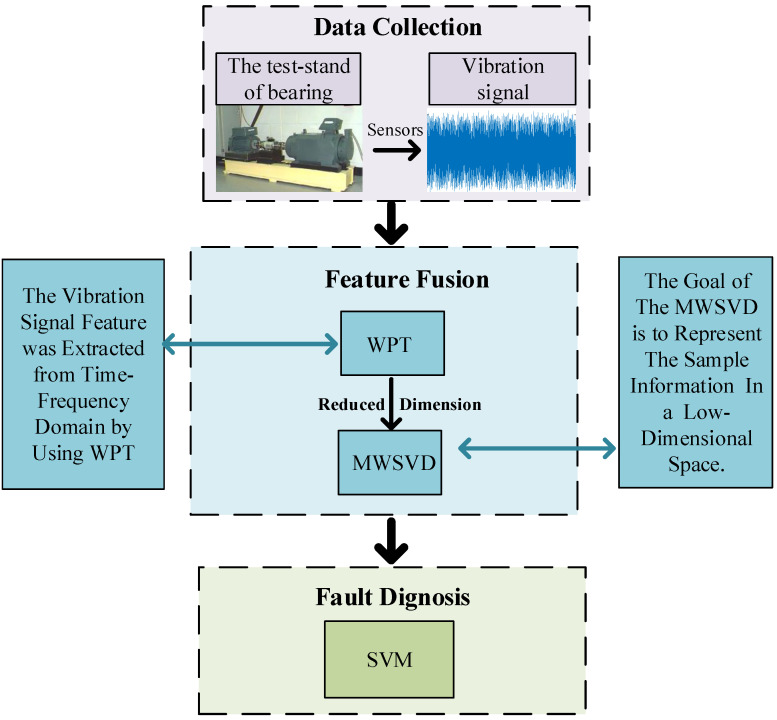
The flowchart of the bearing fault diagnosis method that is based on feature fusion.

**Figure 2 sensors-21-02524-f002:**
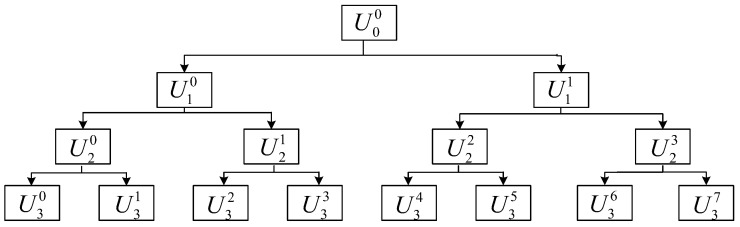
Schematic diagram of the wavelet packet structure decomposition at scale l=3.

**Figure 3 sensors-21-02524-f003:**
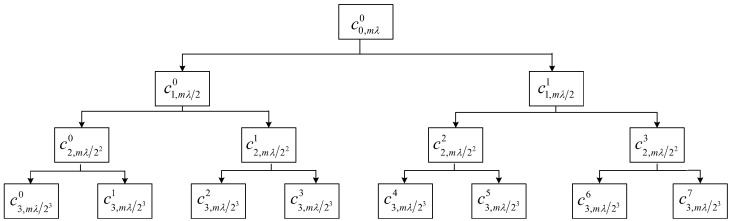
Schematic diagram of Mallat decomposition of wavelet packet at scale l=3.

**Figure 4 sensors-21-02524-f004:**
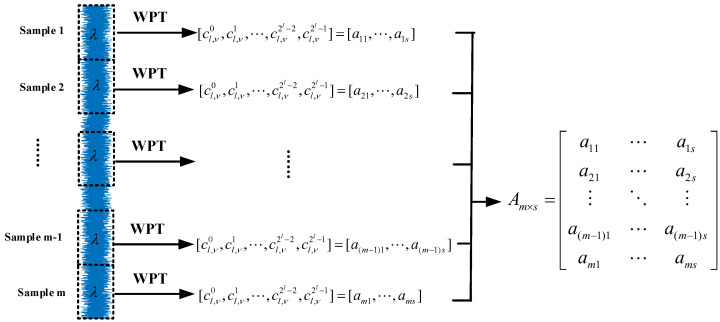
The flowchart of wavelet packet decomposition of bearing signal.

**Figure 5 sensors-21-02524-f005:**
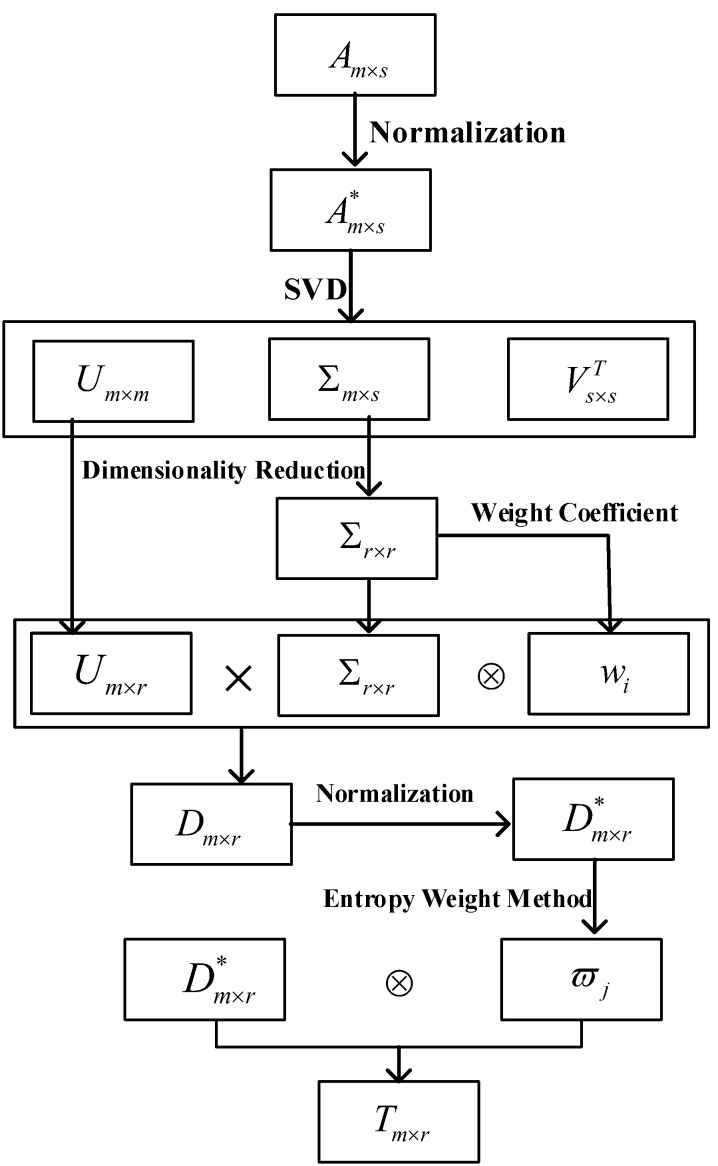
The flowchart of the Multi-Weight Singular Value Decomposition (MWSVD) feature extraction algorithm.

**Figure 6 sensors-21-02524-f006:**
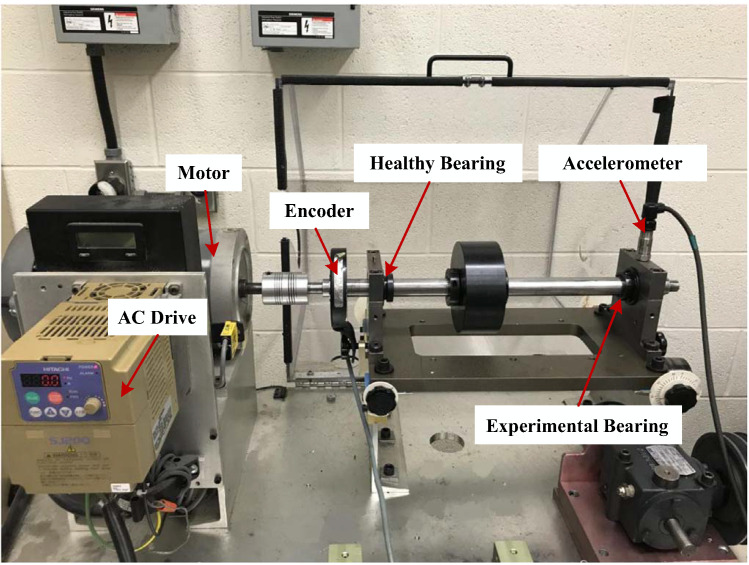
Time-varying bearing experimental device at the University of Ottawa.

**Figure 7 sensors-21-02524-f007:**
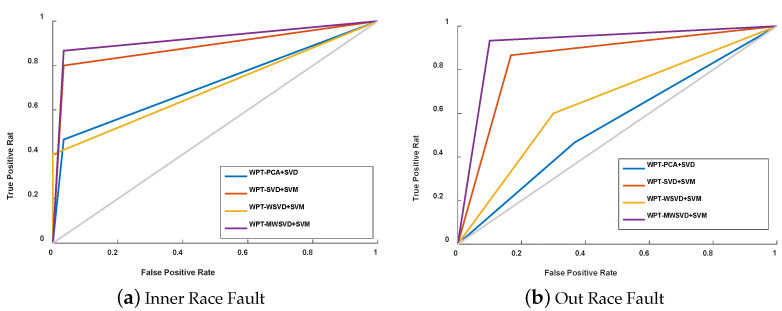
ROC curve diagram of the four algorithms on the fault diagnosis of the bearing inner race (**a**) and outer race (**b**).

**Figure 8 sensors-21-02524-f008:**
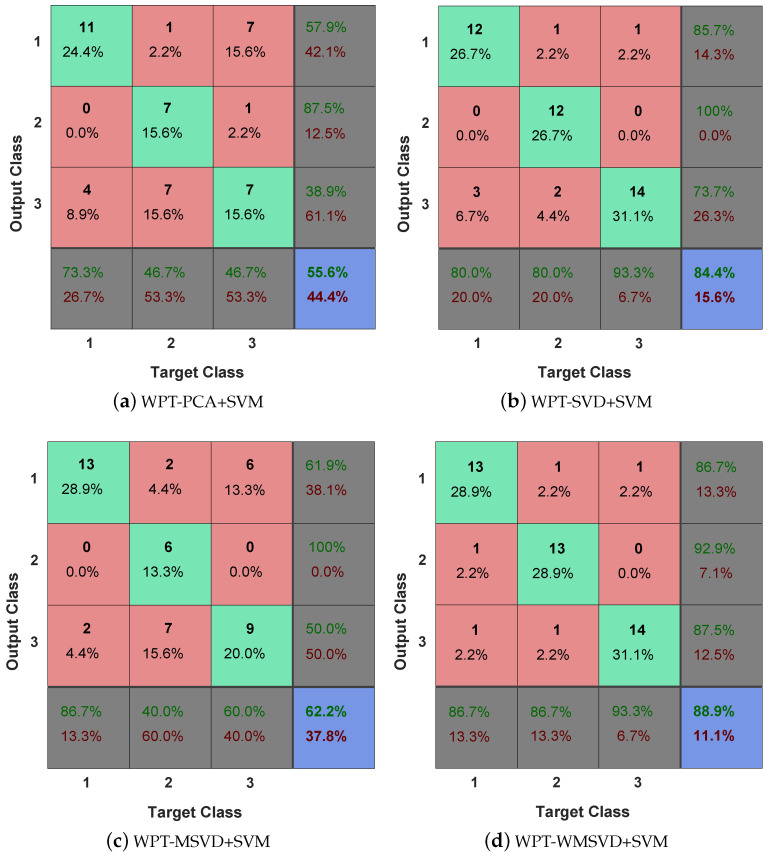
Classification confusion matrix of four algorithms: (**a**) WPT-PCA+SVM; (**b**) WPT-SVD+SVM; (**c**) WPT-MSVD+SVM; and, (**d**) WPT-WMSVD+SVM.

**Figure 9 sensors-21-02524-f009:**
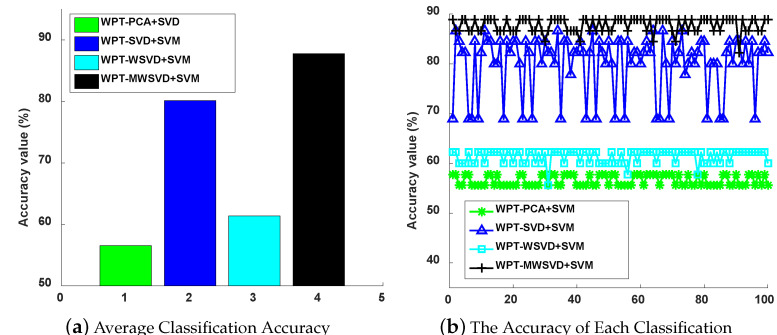
The average classification accuracy of 100 runs (**a**) and each classification accuracy (**b**).

**Figure 10 sensors-21-02524-f010:**
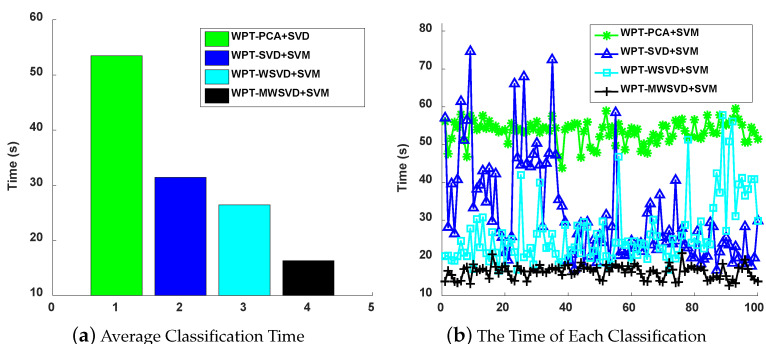
The average classification time of 100 runs (**a**) and each classification time (**b**).

**Figure 11 sensors-21-02524-f011:**
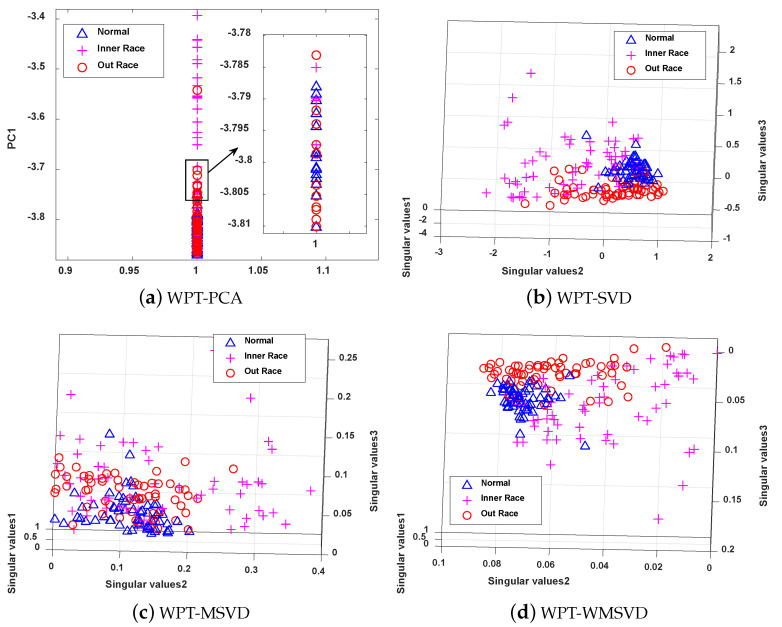
The Visualization of feature extraction effect of four methods: (**a**) Distribution of the data samples based on WPT-PCA; (**b**) Distribution of the data samples based on WPT-SVD; (**c**) Distribution of the data samples based on WPT-WSVD; (**d**) Distribution of the data samples based on WPT-MWSVD.

**Figure 12 sensors-21-02524-f012:**
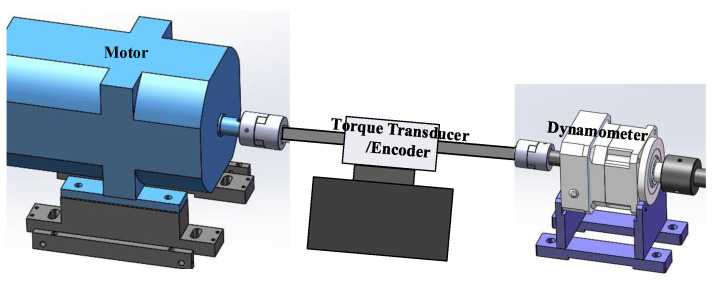
The schematic diagram of bearing experimental platform of Case Western Reserve University.

**Figure 13 sensors-21-02524-f013:**
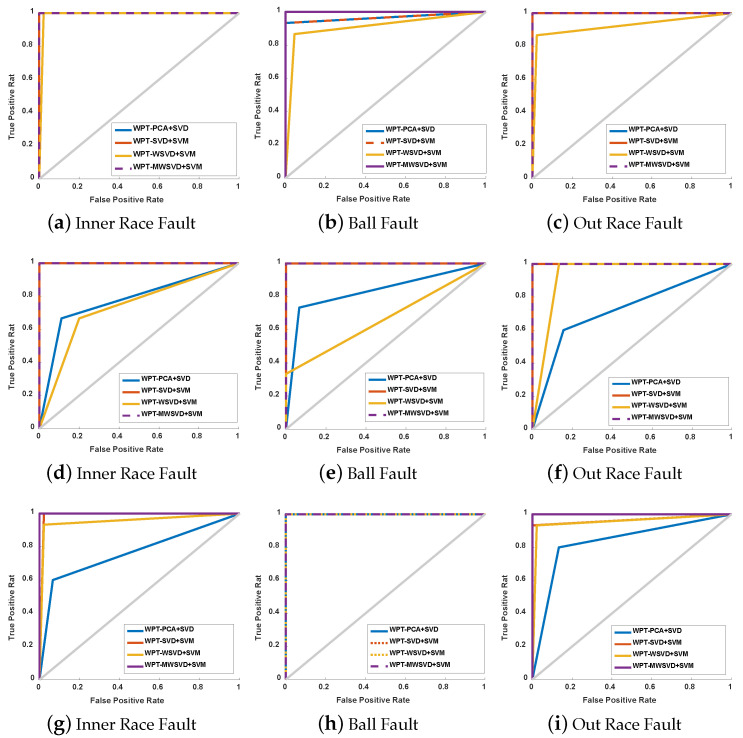
ROC curve diagram of the four algorithms: the minor fault diagnosis of the bearing inner race (**a**), Ball (**b**) and outer race (**c**); the general fault diagnosis of the bearing inner race (**d**), Ball (**e**) and outer race (**f**); the serious fault diagnosis of the bearing inner race (**g**), Ball (**h**), and outer race (**i**).

**Figure 14 sensors-21-02524-f014:**
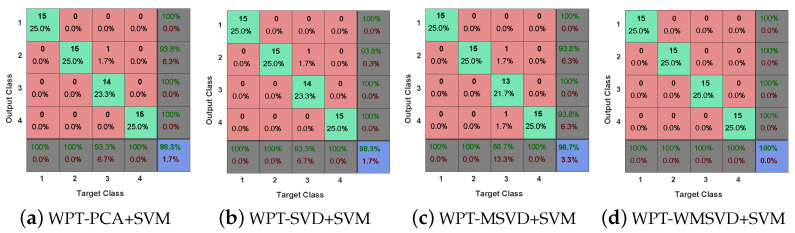
Classification confusion matrix of four algorithms: the minor fault (**a**) WPT-PCA+SVM; (**b**) WPT-SVD+SVM; (**c**) WPT-MSVD+SVM; (**d**) WPT-WMSVD+SVM; the general fault (**e**) WPT-PCA+SVM; (**f**) WPT-SVD+SVM; (**g**) WPT-MSVD+SVM; (**h**) WPT-WMSVD+SVM; the serious fault (**i**) WPT-PCA+SVM; (**j**) WPT-SVD+SVM; (**k**) WPT-MSVD+SVM; and, (**l**) WPT-WMSVD+SVM.

**Figure 15 sensors-21-02524-f015:**
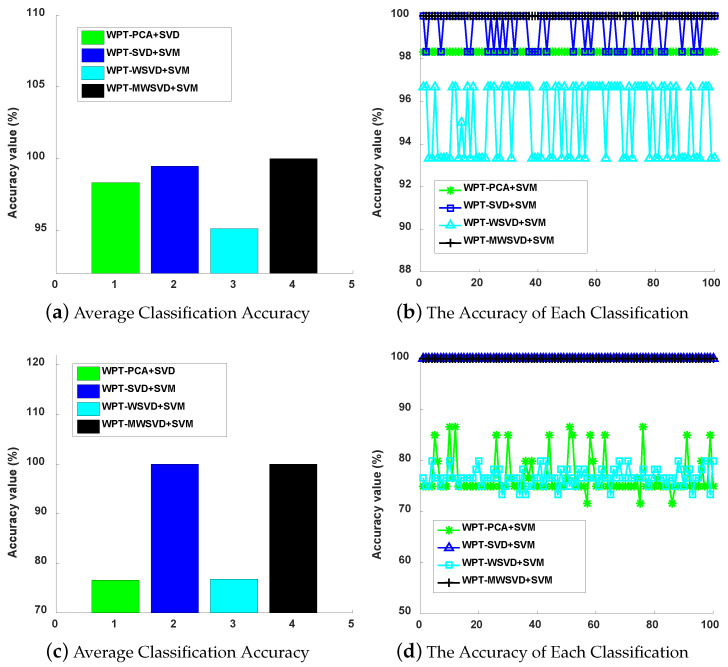
The degree of fault is minor fault (**a**), general fault (**c**) and serious fault (**e**); The degree of fault is minor fault (**b**), general fault (**d**), and serious fault (**f**).

**Figure 16 sensors-21-02524-f016:**
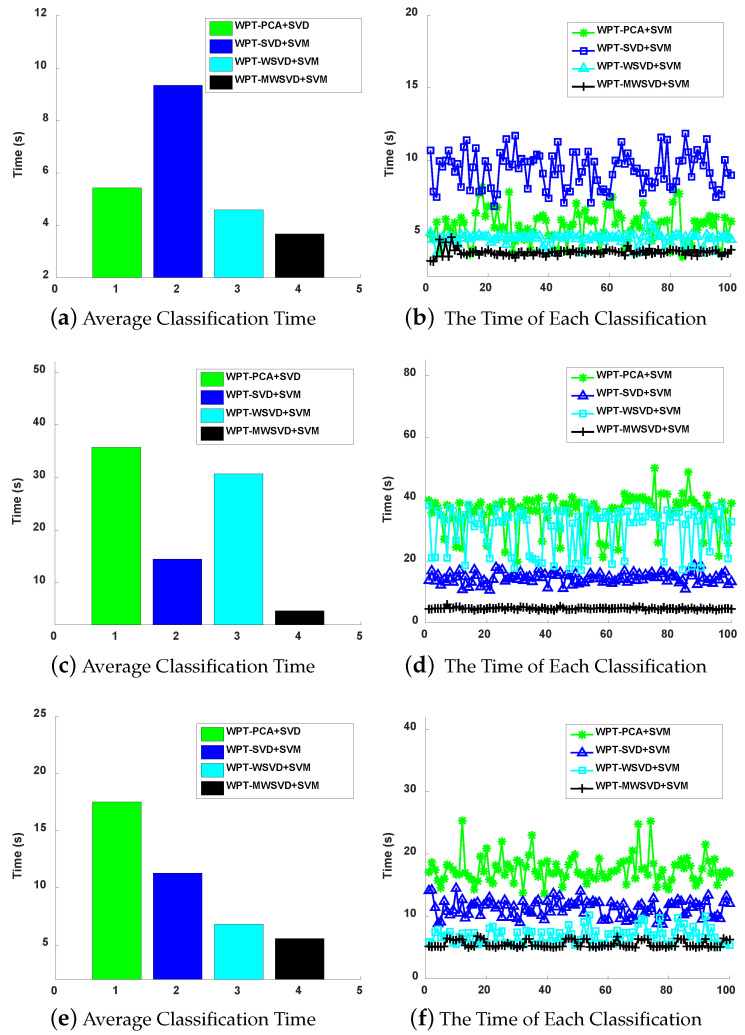
The degree of fault is minor fault (**a**), general fault (**c**) and serious fault (**e**); The degree of fault is minor fault (**b**), general fault (**d**), and serious fault (**f**).

**Figure 17 sensors-21-02524-f017:**
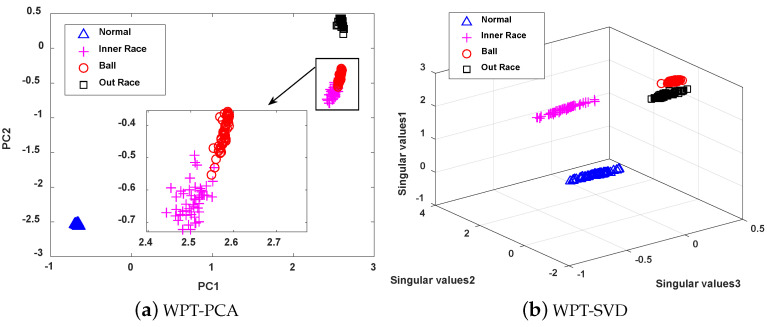
The Visualization of feature extraction effect of four methods under minor faults: (**a**) Distribution of the data samples based on WPT-PCA; (**b**) Distribution of the data samples based on WPT-SVD; (**c**) Distribution of the data samples based on WPT-WSVD; and, (**d**) Distribution of the data samples based on WPT-MWSVD.

**Figure 18 sensors-21-02524-f018:**
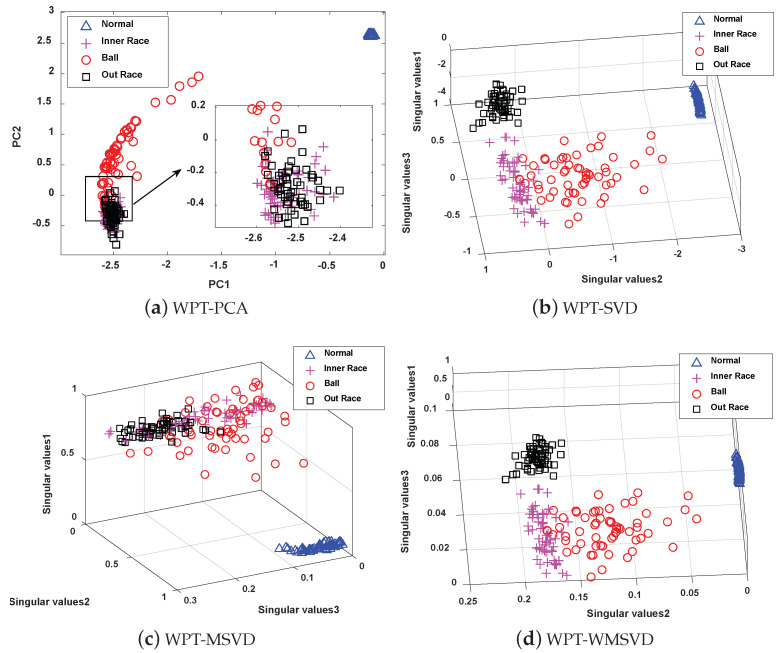
The Visualization of feature extraction effect of four methods under general faults: (**a**) Distribution of the data samples based on WPT-PCA; (**b**) Distribution of the data samples based on WPT-SVD; (**c**) Distribution of the data samples based on WPT-WSVD; and, (**d**) Distribution of the data samples based on WPT-MWSVD.

**Figure 19 sensors-21-02524-f019:**
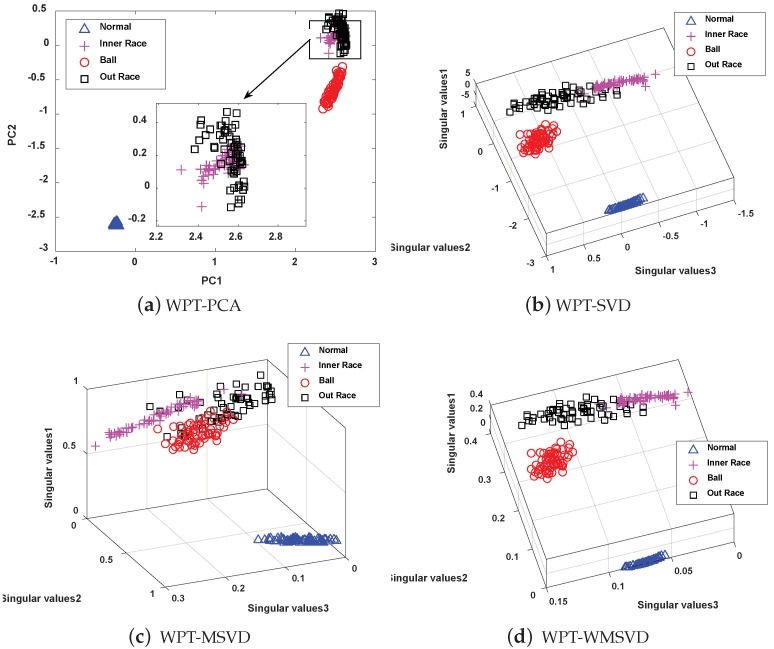
The Visualization of feature extraction effect of four methods under serious faults: (**a**) Distribution of the data samples based on WPT-PCA; (**b**) Distribution of the data samples based on WPT-SVD; (**c**) Distribution of the data samples based on WPT-WSVD; (**d**) Distribution of the data samples based on WPT-MWSVD.

**Table 1 sensors-21-02524-t001:** Each Fault State and Corresponding Label in the Data Set.

Status	Operating Rotational Speed	Label
Normal Status	from 28.6 Hz to 13.9 Hz	1
Inner Race Fault	from 25.8 Hz to 12.0 Hz	2
Out Race Fault	from 25.4 Hz to 10.3 Hz	3

**Table 2 sensors-21-02524-t002:** Experimental Environment and Parameters.

Experimental Environment		Experimental Modal Parameters	
Operating System	Windows 10 Home Chinese Version	The Size of The Sliding Window	λ=1024
CPU	Intel Core i7-8550U @2.00 GHz	Wavelet Function	db10 [40]
RAM	8.00 GB	wavelet packet scale	l=4
System Type	64 bit	The Number of Runs	τ=100
SVM Classifier	LIBSVM by Lin Zhiren of National Taiwan University [41]	Cumulative Contribution Rate of Singular Values	90%

**Table 3 sensors-21-02524-t003:** The optimal parameters of SVM

Fault Diagnosis Model	Penalty Faramenter *C*	Kernel Faramenter ε	CA Accuraey (%)
WPT-PCA+SVM	25.48	990.5	55
WPT-SVD+SVM	5.65	4.13	81.67
WPT-WSVD+SVM	1.19	103.44	66.67
WPT-MWSVD+SVM	25.15	212.47	81.67

**Table 4 sensors-21-02524-t004:** Classification results under different models.

Fault Diagnosis Model	The Average Accuracy η of 100 Runs (%)	Sample Variance of Accuracy δ	The Average Time of 100 Runs (s)	Average Processing Time for 1 s of Data
WPT-PCA+SVM	56.56	1.23	53.46	34.80
WPT-SVD+SVM	80.16	34.04	31.44	20.47
WPT-WSVD+SVM	61.38	1.67	26.43	17.21
**WPT-MWSVD+SVM**	**87.80**	**1.94**	**16.32**	**10.62**

**Table 5 sensors-21-02524-t005:** Failure status label for each degree of fault.

The Severity of the Fault	Fault Size of Samples	Status	Label
Minor Fault	0.1778 (mm)	Normal Status	1
		Inner Race Fault	2
		Ball Fault	3
		Out Race Fault	4
General Fault	0.3556 (mm)	Normal Status	1
		Inner Race Fault	2
		Ball Fault	3
		Out Race Fault	4
Serious Fault	0.5334 (mm)	Normal Status	1
		Inner Race Fault	2
		Ball Fault	3
		Out Race Fault	4

**Table 6 sensors-21-02524-t006:** The optimal parameters of Support Vector Machine (SVM).

The Severity of the Fault	Fault Diagnosis Model	Penalty Faramenter *C*	Kernel Faramenter ε	CA Accuraey (%)
Minor Fault	WPT-PCA+SVM	0.54	89.44	98.33
	WPT-SVD+SVM	1.42	31.46	98.33
	WPT-WSVD+SVM	42.97	4.83	97.50
	WPT-MWSVD+SVM	4.71	691.86	100
General Fault	WPT-PCA+SVM	79.78	39.67	75.42
	WPT-SVD+SVM	0.20	2.32	100
	WPT-WSVD+SVM	5.96	19.02	84.17
	WPT-MWSVD+SVM	4.36	167.58	100
Serious Fault	WPT-PCA+SVM	4.89	34.84	87.08
	WPT-SVD+SVM	4.21	9.62	98.33
	WPT-WSVD+SVM	8.88	57.90	98.75
	WPT-MWSVD+SVM	4.06	623.54	100

**Table 7 sensors-21-02524-t007:** Failure status label for each degree of fault.

The Severity of the Fault	Fault Diagnosis Model	The Average Accuracy η of 100 Runs (%)	Sample Variance of Accuracy δ	The Average Time of 100 Runs (s)	The Average Processing Time for 1 s of Data
Minor Fault	WPT-PCA+SVM	98.33	0	5.43	0.21
	WPT-SVD+SVM	99.48	0.60	9.34	0.36
	WPT-WSVD+SVM	95.11	2.76	4.59	0.18
	**WPT-MWSVD+SVM**	**100**	**0**	**3.67**	**0.14**
General Fault	WPT-PCA+SVM	76.62	13.72	35.72	1.38
	WPT-SVD+SVM	100	0	14.50	0.57
	WPT-WSVD+SVM	76.78	2.93	30.66	1.20
	**WPT-MWSVD+SVM**	**100**	**0**	**4.67**	**0.18**
Serious Fault	WPT-PCA+SVM	85.85	0.76	17.52	0.68
	WPT-SVD+SVM	98.53	0.30	11.29	0.44
	WPT-WSVD+SVM	97.73	0.65	6.82	0.27
	**WPT-MWSVD+SVM**	**99.52**	**0.58**	**5.57**	**0.22**

## Data Availability

Not applicable.

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
