# Peer review of "Bearing Fault Feature Extraction and Fault Diagnosis Method Based on Feature Fusion"

_sensors, 2021, doi:10.3390/s21072524_

Round 1
Reviewer 1 Report
This paper provided a feature extraction technique for bearing faults diagnosis. Several issues need to be addressed and revised before further consideration.
- The authors claimed to develop a novel feature extraction/dimension reduction technique for fault diagnosis, which could outperform the traditional feature engineering techniques. However, the related works are not extensively discussed. Traditional feature extraction techniques for feature extraction, such as auto-encoder or deep neural networks are not discussed. Please indicate that why the SVD-based techniques are selected over the rest of the feature extraction techniques. Papers you may refer to: Fault diagnosis for rotating machinery using multiple sensors and convolutional neural networks; Unsupervised Cross-domain Fault Diagnosis Using Feature Representation Alignment Networks for Rotating Machinery.
- It seems the description in lines 99-108 is not consistent with Figure 1. In the text description, the vibration signal is passed to WPT to get the time-frequency domain feature, then the dimension of the time-frequency domain feature is reduced by MWSVD, which is processed sequentially. However, in Figure 1, WPT and MWSVD seem to run parallelly.
- Please why using SVM as the classifier, not other linear/nonlinear classifiers? Please discuss and compare.
- Please discuss and compare the difference between WSVD and MWSVD. Perhaps a preliminary study would help the readers to better understand the contribution of this paper.
- The font size in some figures is very small (e.g. Figure 6).
- As both WSVD and MWSVD developed from SVD, please also compare the result of SVD (i.e., WPT-SVD+SVM) in the experiments.
- Please provide the axis label for the left-most image of Figure 9. And explain why there is an inner plot for features. Besides, this sub-figure is simply presented but not discussed. As for the rest of the subfigures in Figure 9, Please present them in the 3D form as they have 3 SVD dimensions. Current figures seem to be in 2D.
- The axis range in Figure 12 is incorrect, which does not present the result of MWSVD. And its font size is also very small.
- Please discuss why the variance of accuracy for MWSVD is consistently to be 0 in Table 3 and Table 5. It would be best if the authors could explain which part of the proposed network helped to reduce the variance significantly.
- Please explain why MWSVD is more efficient and requires less computing time than the benchmark algorithms.
- In Table 5, the proposed method, MWSVD, is not the best algorithm in both minor faults and serious faults scenarios. In contrast, the PCA and WSVD performed badly in the general fault case. Please explain why MWSVD performed poorly in minor/serious faults scenarios, and why the accuracy of PCA and WSVD drop significantly in the general fault case.
Reviewer 2 Report
Dear Authors,
Below you can read my comments on your article.
Sincerely yours.
Reference: sensors-1125892
Title: Bearing Fault Feature Extraction and Fault Diagnosis Method Based on Feature Fusion
Authors: Huibin Zhu, Zhangming He, Juhui Wei, Jiongqi Wang, Haiyin Zhou
Overview and General Recommendation
Bearing diagnosis is still of interest for academics and industry. Many papers are published recently about this topic. The problem is of interest because bearing fault detection is still challenging for different reasons: energy of fault features is very low compared to other components present in signals; noise is usually high and buries fault features; feature extraction from signals is difficult; bearing failure modes are diverse; quality of the signals depends strongly on the sensor and its location, such as in vibration measurements.
Research in this area focuses on advanced and powerful signal processing techniques, feature extraction tools, feature fusion approaches, and advanced classification algorithms based on machine learning or artificial intelligence.
The authors propose a diagnosis methodology based on feature extraction and fusion methodology. Wavelet Packet Transform is used for feature extraction, but there is no novelty here. The feature fusion method is named Multi-weight Singular Value Decomposition (MWSDV). This may be the main contribution of this paper. SVM is used for fault classification, which is also not new, and the data comes from two repositories.
The article needs much improvement, especially the introduction and the results section. Therefore, I recommend that a major revision is warranted. I explain my concerns in more detail below. I ask that the authors specifically address each of my comments in their response.
Major comments
- In my opinion, the introduction section is too long. I recommend the authors to avoid the use of ambiguous words and terms such as "etc." Please, be more specific. There is a sentence that troubles me: "However, the bearing vibration signals often show strong nonlinear and non-stationary characteristics [15], which makes it difficult for traditional signal analysis methods based on the time domain or frequency domain to effectively extract bearing signal features." In my experience with industry and lab works, these all depend on the used sensor, its quality, its location or position, the specific bearing fault. and on the machine operation. The word "often" is not adequate.
- Sections 2 and 3 are well written and offer the reader a detailed explanation of the proposed methodology's mathematics. But section 2 needs some improvements and error corrections:
- Mallat is not cited in the text. His algorithms are used, but cites are missing.
- "when the sampling interval is sufficiently small ..." Please, be more specific. This explanation is too vague. What is it compared to?
- Line 160, page 6. An ambiguous term is used again: "quite".
- From line 163. I think there is a notation problem. The normalized feature matrix is named with superscript "*", but sometimes this is missing, making it difficult to follow the explanation of algorithm number 2. I may be wrong, but I would appreciate it if the notation is revised.
- Section 4. Experiments and results. In my opinion, this is the section that needs much improvement.
- As two different data sets are used, I would start the section with their description. Sampling frequencies are very different. The number of data points used is the same, but this means a big disparity in the time length of data samples. In my opinion, the authors should analyze if differences between datasets affect the results, or it is an advantage to prove the effectiveness of their proposal.
- In my opinion, figures are not informative for different reasons: too small; x labels missing; not appropriate for making comparisons.
- My suggestion is to summarize important numeric information in tables.
- Classification performance is better analyzed using ROC curves. The reader can analyze in a single glance how the different algorithms are classifying, and the comparison is straightforward. The ROC curves of the three diagnosis methods must be in the same figure.
- Confusion matrices are also very useful as they provide qualitative information about the classification results. In my opinion, accuracy is not as important as to know where the classifier is failing.
- Figures 9, 13, 14 and 15 are important to illustrate the effectiveness of the feature fusion and extraction method. But, in their actual size are useless.
Strengths of the article
It is demonstrated that the methodology proposal improves classification performance. The authors point out in the introduction the novel contributions very well. The contents and contributions of the paper are adequately summarized in the last section.
Reviewer 3 Report
-English should be corrected
Line 280 for example "To cope with the prolem that it"
-please add colorful picture of measurements (optionally);;; + arrows what is what
-please add block diagram of the proposed research step by step ;;; what is the result of paper?;;;
-please add block diagram of the proposed method;;;;
-please add photo/photos of application of the proposed research ;;;;
-please add sentences about future analysis;;;
-Figures should have better quality;;;; for now they are low
-Fonts of figures should be bigger;;;
-Please add labels to axes (Figures);;;;
-please add arrows to photos what is what;;;
-formulas and fonts should be formatted;;;;
-references should be 2018-2021 Web of Science about 50% or more ;; 30 at least
-Please compare with other methods, justify. Advantages or Disadvantages different methods
for example:
1) Acoustic fault analysis of three commutator motors, Mechanical Systems and Signal Processing, vol. 133 art. no. 106226, 2019,
https://doi.org/10.1016/j.ymssp.2019.07.007
-Conclusion: point out what are you done;;;;
-is there possibility to use the proposed method for other problems?
Reviewer 4 Report
-This work lies in the proposal of diagnosis methodology based on a "Feature Fusion" strategy to diagnose different faulty conditions on bearings.
-The proposal lacks of novelty and several issues must be addressed for being improved.
-The introduction is well described, nevertheless, the discussion of the state of the art (SoA) works is focused only to works that consider SVM or WPT techniques. Thus, the discussion of SoA may be improved.
-In most of the reported researches, the "Feature Fusion" is reported as the fusion of features that are extracted from different physical magnitudes or features that are estimated from different domains (time, frequency and time-frequency). In this regard, this proposal is also based on a "Feature fusion" approach, however, it is not clear how features are fused.
-The WPT is used to estimated a meaningful set of features to characterise vibrations signals that belong to different bearing conditions. What are the advantages of the WPT respect to other techniques such as EMD?
-In Figure 14 is presented a comparison of methods where a visualization of the feature extraction effects may be appreciated. Why the visualization of the WPT-PCA (Figure 14) is represented into a 2-dimensional space? whereas, the WPT-MWSVD feature extraction is presented into a 3-dimensional space?. If it is intended to make a comparison, it should be done under the same considerations (same dimensional space).
-A description of the experimental test benches is missing.
-The parameters that have been used in the SVM are also missing.
-There are several grammar mistakes, i. e., "prolem" in section 5 "Conclusion"
Round 2
Reviewer 1 Report
The authors addressed some of my previous comments. However, a few key issues should be addressed before further consideration.
- I am not satisfied with the explanation of my previous comment on MWSVD's efficiency. I could not understand why "the role of the model is far greater than the non-sensitive features" results in a more efficient model. Please explain.
- The results in Tables 3 and 5, Figures 13 and 14 suggest that the proposed method cannot outperform the benchmark algorithm significantly. I agree with reviewer 2 that the authors should state the importance and necessity of the proposed method.
- The organization of the manuscript should be improved. For example, the added description for WSVD in Section 2.2 should be placed before the methodology section as it is not proposed by the authors.
Reviewer 2 Report
Reference: sensors-1125892 (Second revision)
Title: Bearing Fault Feature Extraction and Fault Diagnosis Method Based on Feature Fusion
Authors: Huibin Zhu, Zhangming He, Juhui Wei, Jiongqi Wang, Haiyin Zhou
Dear Authors,
I really appreciate the work done to improve the paper. I know it is not easy to attend to three reviewers' requirements and on such short notice.
I believe that you did remarkable research work. Your idea is excellent, but I still doubt about your algorithm's superiority compared to the other three. I will try to explain myself next.
ROC curves in Figure 7 show that:
- WPT-MWSVD+SVM is as good as WPT-WSVD+SVM and WPT+SVD+SV in classifying the inner race fault. This means that MWSVD does not improve classification results compared to other methods.
- WPT-MWSVD+SVM is as good as WPT-WSVD+SVM in classifying the outer race fault. Both methods are better than the other two methods.
I am afraid that MWSVD does not improve the results compared to WSVD. Confusion matrices in figure 8 confirm these results, where (c) and (d) are identical. The authors must revise these results because the sums by rows should be identical in the four matrices, and they are not.
I'm afraid I have to disagree with the authors claim in lines 209 to 212. Figures 9 and 10, and table 3 demonstrate that WPT-MWSVD+SVM is only slightly better than WPT-WSVD+SVM.
Next, it is a summary of the ROC curves shown in figure 13:
- Inner Race fault (A1): WPT- MWSVD+SVM is as good as WPT-SVD+SVM. All methods easily detect this fault.
- Ball fault (B2): WPT- MWSVD+SVM is as good as WPT-SVD+SVM.
- Out Race fault (C1): WPT- MWSVD+SVM is as good as WPT-SVD+SVM and WPT-PCA+SVM.
- Inner Race fault (A2): WPT- MWSVD+SVM is as good as WPT-SVD+SVM. Both methods are much better than the other two.
- Ball fault (B2): WPT- MWSVD+SVM is second good behind WPT-SVD+SVM.
- Out Race fault (C2): WPT- MWSVD+SVM is as good as WPT-SVD+SVM. The performance of the other two is much worse.
- Inner Race fault (A3): WPT-WSVD+SVM obtains the best classification performance. WPT- MWSVD+SVM is as good as WPT-SVD+SVM, and both are a little behind WPT-WSVD+SVM.
- Ball fault (B3). All algorithms show the same performance. This fault is easily classified.
- Out Race fault (C3): WPT-WSVD+SVM achieves the best performance. WPT- MWSVD+SVM is as good as WPT-SVD+SVM, and both are a little behind WPT-WSVD+SVM.
Confusion matrices in figure 14, and figure 15 are also not conclusive about the superiority of WPT- MWSVD+SVM compared to the other three algorithms. The only sure conclusion is that PCA gives worse diagnosing performance compared to the other three.
Figure 16 shows that the proposed method is faster than the other three algorithms regarding the classification time, but I am not sure if this is important.
Figure 17 is very illustrative for comparison purposes. There is no apparent difference between WPT+MWSVD and WPT-SVD. I believe that both algorithms are equally effective in separating features under minor faults. After observing figures 18 and 19, I come to the same judgment.
Therefore, my conclusion is that MWSVD is not superior to WSVD. The authors have demonstrated that the classification time is improved, but my questions are: Is this important?; and Why should it be necessary?
I will not ask the authors for any further changes. The article is fine as it is now. However, the authors should answer the two questions above and modify the introduction and conclusion sections accordingly to convince the reader that their proposed algorithm is necessary and makes a valuable contribution to the state of the art. In my humble opinion, after reading this second version, that is not clear.
Sincerely yours.
Reviewer 3 Report
The authors didnt compare with SVM approaches for example acoustic signals.
What are advantages and disadvantages acoustic analysis, vibration analysis, please compare
Recognition of Acoustic Signals of Loaded Synchronous Motor Using FFT, MSAF-5 and LSVM, Archives of Acoustics, 2015, 40 (2), pp. 197-203.
Figure 12 I saw it 30 times, please add another photo.
https://doi.org/10.1515/aoa-2015-0022
Reviewer 4 Report
The comments and suggestions made in the last revision have been attended, however, the following issues remains and should be properly addressed:
- A brief description of the advantages of WPT over the EMD has to be included.
- In Figures 17, 18, 19, the projection obtained by applying WPT-PCA may be presented also into a 3D-space. (There in no a clear justification about why it is presented in 2D)
- If the parameters used in the SVM have been tuned through a genetic algorithm, results obtained about this process should be also included (i.e., the convergence performance), also, it should be discussed how the parameter setting affects the results.
Round 3
Reviewer 1 Report
The authors have addressed my previous comments, I have no further questions.
Author Response
Thanks again to the reviewers for their tremendous efforts for our manuscript.
Reviewer 2 Report
Reference: sensors-1125892 (Second revision)
Title: Bearing Fault Feature Extraction and Fault Diagnosis Method Based on Feature Fusion
Authors: Huibin Zhu, Zhangming He, Juhui Wei, Jiongqi Wang, Haiyin Zhou
Dear Authors,
I think you have done much work to improve the results presented in the paper. You have even made changes to the algorithm proposed in section 3.2. I also accept your point about the importance of computational speed.
However, after re-reading the article, I noticed some worrying changes to the results in section 5.1. The changes made to the algorithm should improve the classification efficiency of WPT-MWSVD+SVM but should not affect the results of the other algorithms. Figures 7, 8, and 9 of the second and third versions show a dramatic worsening of the other three algorithms' classification results. After re-reading the document, I do not understand these changes. Please explain why this is not the case in section 5.2 where figures 9, 10, and tables 4 (v3) and 3 (v2) are not so different.
Although the editorial staff will notice, please revise the manuscript as I noticed some words in your mother tongue, as in line 208.
Sincerely yours.
Reviewer 3 Report
-fonts of figures should be bigger
Author Response
The author processed the pictures of the article in accordance with the recommendations of the reviewers. Thanks again to the reviewers for their tremendous efforts for our manuscript.
Round 4
Reviewer 2 Report
Dear Authors,
I still don't understand the differences between the classification results of case 1 between version 1 and 2 of the manuscript. The new sentence does not help me to understand it. If given the chance, my recommendation is to improve that explanation.
Best regards,
Daniel
Author Response
Please see the attachment

This manuscript is a resubmission of an earlier submission. The following is a list of the peer review reports and author responses from that submission.
Round 1
Reviewer 1 Report
Dear Authors,
Regarding the first round review of this manuscript, the reviewer has the following comments:
- Introduction: the contribution is not clear. So, please explain about the contributions in more details.
- Introduction: Please add a block-diagram regarding the proposed algorithm and explain about it and sub-blocks deeply.
- Results: Please explain in more details about the test scenarios.
- How the authors can validate the robustness and stability in the proposed algorithm?
- As you know that reliability is one of the main challenge, so how you can validate reliability in your technique?
- Method: Why SVM is selected for classification?
- How about the results, if the authors used deep-learning method instead of machine learning based method (e.g., SVM)?
Regards,
Reviewer 2 Report
The manuscript presents a methodology for failure detection on rolling bearings using vibration signals.
On the introduction the problem is sufficiently commented and previous experience are clearly identified. On the introduction the authors present the Wavelet packet transform (WPT) as a proper methodology to analyze non stationary signals. MultipleWeighted Singular Value Decomposition method (MWSVD) is used as a fault indicator, based on entropy analysis. Finally the authors propose a support vector machine to finally classify the fault condition. The introduction presents some of the limitations of previously studied options to remove irrelevant and redundant features for the fault detection. PCA, SVD and WSVD are adequately introduced.
On section 2 the MWSVD is correctly presented. Nevertheless, even the presented methodology involving three different steps (SVD, Dimensionality Reduction and Entropy weight) is detailed the wavelet decomposition used is not sufficiently justified. Why does the authors considers Daubechies 4?
Equally the Support Vector Machine (SVM) is presented on section 3. The theoretical approach on the SVM is correctly introduced, nevertheless the labelling process is not sufficiently described. Authors should provide more information on how the labelling is done.
The two case studies are conveniently selected, both of them take measurements on steady state, so the presented methodology does not takes an actual advantage on the wavelet transform. An accurate comparison should be done with standard fourier and filtering methods that can be easily used on steady state.
The authors should introduce some details on the computational cost on the presented methodology and the actual advantages on standard industrial approach based on FFT used for fault detection. Finally, considering an actual application the authors could discuss how to approach noisy signals coming from an industrial environment.
Reviewer 3 Report
The submitted manuscript fundamentally touches two very important points - first is that it concerns a very important group of components (bearings) that account for significant rotating machines failures and second is that it deals with data fusion. The article is well-constructed and contains very valuable information that would enrich current knowledge on rotating machines faults diagnosis. While the study makes reference to several articles in the areas of faults diagnosis, data fusion and machine learning, however, I do feel that the distruction of the cited articles is skewed and not all-encompassing. For instance, there were no mention of spectrum based data and feature fusion approaches such as composite spectra, coherent composite spectra and their higher ordeer equivalents which immensely reduce number of sensors and data required. Also, there should be inclusion of more feature fusion techniques that can handle nonlinear data sets such KPCA. May I suggest that the authors include the following articles and others:
Liu QC, Wang HP. A case study on multisensor data fusion for imbalance diagnosis of rotating machinery. Artificial Intelligence for Engineering Design, Analysis and Manufacturing: AI EDAM. 2001 Jun 1;15(3):203.
Yunusa-Kaltungo A, Sinha JK, Elbhbah K. HOS analysis of measured vibration data on rotating machines with different simulated faults. InAdvances in condition monitoring of machinery in non-stationary operations 2014 (pp. 81-89). Springer, Berlin, Heidelberg.
Azamfar M, Singh J, Bravo-Imaz I, Lee J. Multisensor data fusion for gearbox fault diagnosis using 2-D convolutional neural network and motor current signature analysis. Mechanical Systems and Signal Processing. 2020 Oct 1;144:106861.
Wang Z, Xiao F. An improved multisensor data fusion method and its application in fault diagnosis. IEEE Access. 2018 Dec 24;7:3928-37.
Jafari H, Poshtan J, Sadeghi H. Application of fuzzy data fusion theory in fault diagnosis of rotating machinery. Proceedings of the Institution of Mechanical Engineers, Part I: Journal of Systems and Control Engineering. 2018 Aug;232(8):1015-24.
Hu X, Xiao Z, Liu D, Tang Y, Malik OP, Xia X. KPCA and AE Based Local-Global Feature Extraction Method for Vibration Signals of Rotating Machinery. Mathematical Problems in Engineering. 2020 Jun 29;2020.
Yunusa-Kaltungo A, Sinha JK. Effective vibration-based condition monitoring (eVCM) of rotating machines. Journal of Quality in Maintenance Engineering. 2017 Aug 14.
The theoretical background of the techniques deployed in the paper were adequately demonstrated.
Once the above highlighted points are addressed, the paper can accepted for publication.
Reviewer 4 Report
The paper should be improved. The authors should compare the results of the research with the recent research (2020). At this time, the authors cannot prove the originality of the research.
The fonts of the figures title should be increased.
The Conclusion section should start with the capital character.
Round 2
Reviewer 1 Report
Dear Authors,
Thank you for your response letter. Regarding the second round review the reviewer believes that this manuscript can be accepted for further processing.
Regards,
Reviewer 2 Report
The authors improved the manuscript, the comments and improvements are clearly presented. Nevertheless, there is an important weakness on the proposal.
Time frequency analysis and wavelet analysis are a great solution for transient state analysis where standard fourier analysis presents a challenge due to windowing. But, for steady state, as it is presented the methodology is costly and lacks of actual interest.
My suggestion is, since the authors introduce on their resposne that some transient experiments are on the way, present a great manuscript introducing the case of transient state operation,
Reviewer 4 Report
The paper has been improved by the authors. In my opinion, the contribution is clearly explained and has been verified both experimentally and by simulation.